# Peroxide-Induced Synthesis of Maleic Anhydride-Grafted Poly(butylene succinate) and Its Compatibilizing Effect on Poly(butylene succinate)/Pistachio Shell Flour Composites

**DOI:** 10.3390/molecules26195927

**Published:** 2021-09-30

**Authors:** Sandra Rojas-Lema, Jordi Arevalo, Jaume Gomez-Caturla, Daniel Garcia-Garcia, Sergio Torres-Giner

**Affiliations:** 1Technological Institute of Materials (ITM), Universitat Politècnica de València (UPV), Plaza Ferrándiz y Carbonell 1, 03801 Alcoy, Spain; jorarag1@epsa.upv.es (J.A.); jaugoca@epsa.upv.es (J.G.-C.); dagarga4@epsa.upv.es (D.G.-G.); 2Research Institute of Food Engineering for Development (IIAD), Universitat Politècnica de València (UPV), Camino de Vera s/n, 46022 Valencia, Spain

**Keywords:** biopolymers, green composites, reactive compatibilization, food waste valorization, Circular Bioeconomy

## Abstract

Framing the Circular Bioeconomy, the use of reactive compatibilizers was applied in order to increase the interfacial adhesion and, hence, the physical properties and applications of green composites based on biopolymers and food waste derived lignocellulosic fillers. In this study, poly(butylene succinate) grafted with maleic anhydride (PBS-*g*-MAH) was successfully synthetized by a reactive melt-mixing process using poly(butylene succinate) (PBS) and maleic anhydride (MAH) that was induced with dicumyl peroxide (DCP) as a radical initiator and based on the formation of macroradicals derived from the hydrogen abstraction of the biopolymer backbone. Then, PBS-*g*-MAH was used as reactive compatibilizer for PBS filled with different contents of pistachio shell flour (PSF) during melt extrusion. As confirmed by Fourier transform infrared (FTIR), PBS-*g*-MAH acted as a bridge between the two composite phases since it was readily soluble in PBS and could successfully form new esters by reaction of its multiple MAH groups with the hydroxyl (–OH) groups present in cellulose or lignin of PSF and the end ones in PBS. The resultant compatibilized green composites were, thereafter, shaped by injection molding into 4-mm thick pieces with a wood-like color. Results showed significant increases in the mechanical and thermomechanical rigidity and hardness, meanwhile variations on the thermal stability were negligible. The enhancement observed was related to the good dispersion and the improved filler-matrix interfacial interactions achieved by PBS-*g*-MAH and also to the PSF nucleating effect that increased the PBS’s crystallinity. Furthermore, water uptake of the pieces progressively increased as a function of the filler content, whereas the disintegration in controlled compost soil was limited due to their large thickness.

## 1. Introduction

The rising concern toward environmental issues has stimulated research efforts in the perspective of reducing plastic waste given the fact that, in recent decades, the disposal and non-biodegradability of petroleum derived plastics have caused serious water and land pollution issues. In this context, fully or partially bio-based and biodegradable polymers currently represent the most promising options to replace petrochemical polymers. Some of the biopolymer matrices commonly used are biopolyesters, which mainly include polyhydroxyalkanoates (PHAs) and polylactides (PLAs) as well as aliphatic polyesters that are fully or partially obtained from petroleum but are biodegradable, such as poly(ε-caprolactone) (PCL), poly(butylene succinate) (PBS), poly(butylene succinate-*co*-butylene adipate) (PBSA), polybutylene(adipate-*co*-terephthalate) (PBAT), etc. [1]. Among them, PBS is a semicrystalline polyester that can naturally degrade in industrial composting facilities and, under certain conditions, in natural environmental conditions due to the action of some bacteria and fungi [2]. PBS is currently produced at the industrial level by condensation polymerization, that is, polycondensation of petroleum derived 1,4-butanediol and succinic acid. However, both the diacid and diol monomers are underdevelopment to be obtained by the bacterial fermentation route from renewable resources [3]. In general terms, PBS presents good processability and balanced mechanical and thermal properties, closely comparable to some polyolefins [4,5]. In particular, it shows tensile and impact strengths similar to those of polypropylene (PP), whereas its glass transition temperature (T_g_) and melting temperature (T_m_) are approximately −32 and 115 °C, respectively, resulting in a thermal profile similar than low-density polyethylene (LDPE) [6]. Furthermore, this biopolyester has a wide melt-processing window, which makes it suitable for extrusion, film blowing, injection molding, fiber spinning, or thermoforming, finding potential applications as packaging films, office supplies, clothing, bags for compost, vegetation nets, mulching films [7,8] and also in biomedical applications [9,10]. Nevertheless, PBS is still currently considered an expensive biopolymer, with prices in the 3–6 USD/kg range, which is considered to be the main limitation for applications of products fully based on this biopolyester [11].

In this regard, the use cost-effective lignocellulosic fillers could facilitate the market penetration of PBS into the plastics industry in the form of green composites. Thus, the most interesting options are related to the valorization of food or agricultural residues since these raw materials are virtually free and can further improve biodegradability, contributing to the development of the so-called Circular Bioeconomy [12]. As a result, a wide range of agricultural and food waste derived fillers have been recently incorporated into biopolymers in the form of particles or fibers, for example, almond shell [13], coconut fibers [13], orange peel [14], hazelnut shell [15], or coffee husk [16]. In the frame of the Circular Bioeconomy, significant efforts have been particularly dedicated to promote nutshells as lignocellulosic fillers for green composites [17]. In this context, “pistachio” (*Pistacia vera* L.), a genus that belongs to the *Anacardiaceae*, is a nut cultivated and available mostly in The United States, Middle East, and some Mediterranean countries [18,19]. In recent years, the annual production of pistachio has increased, especially in Turkey and Iran, reaching a global production of 638,000 t [20]. The pistachio fruit can be classified as a semidry drupe and it is characterized by a high nutritional value, unique flavor, and a bright green color under a purplish skin [21]. However, the seed (kernel) is encased by a thin soft and clear brown coat (testa) enclosed by a creamy lignified shell (endocarp) that is surrounded by a green to yellow-red colored fleshy hull (mesocarp and epicarp), which are not edible and constitute approximately 50 wt.% of the pistachio nut [22]. Both the hulls and hard shells are currently considered waste from the pistachio industry and, therefore, become a great source of residue or by-product of low economic value. Besides, in the nut trade, it must be considered that pistachio has the advantage that is unique because of shell splitting naturally before harvesting, offering an economical and sustainable advantage compared to almonds or walnuts where their kernels require to be separated from their shells by mechanical cracking [23].

The current major applications of pistachio shell are related to animal feeding and general uses as biomass [24,25], being more recently explored as a source energy [26]. However, interestingly, pistachio shell particles present high hardness since it is mainly composed by cellulose (~31 wt.%), hemicellulose (~42 wt.%), lignin (~21 wt.%), crude protein (~1 wt.%), and ashes (~1 wt.%) [27], so these can be ground to produce the so-called pistachio shell flour (PSF) that can be used to prepare polymer composites with higher mechanical strength and hardness. For instance, Gürü et al. [28] investigated the possibility of using PSF particles with urea-formaldehyde in composites of particleboard with improved hardness and fire retardant properties. In other study, powder of pistachio shell was mixed with a polyester resin (Polipol 3401-TAB) in order to obtain a polymer composite [29], showing that low filler content of 5 and 10 wt.% allows to achieve a better dispersion of the particles and higher mechanical properties. Later, in the research performed by Najafabadi et al. [30], high-density polyethylene (HDPE) was filled with PSF and nanoclays (Cloisite 20A) to improve the mechanical resistance of the polyolefin. More recently, Karaağaç [31] reported the addition of pistachio shell powder as filler in a rubber matrix composed of natural rubber (SIR 20) and styrene-butadiene (CAROM 1502), improving the abrasion resistance, thermal stability, and tensile and aging properties. Nevertheless, this residue has been barely used as raw material in combination with biopolymers to yield green composites.

Unfortunately, the incorporation of lignocellulosic fillers into polymers habitually leads to several issues due to the polar hydroxyl (−OH) groups present on the surface of cellulose having difficulty in forming a well-bonded interface with the non-polar polymer/biopolymer matrix since the hydrogen bonds tend to prevent the wetting of the filler surfaces. This low compatibility between both composite components then affects the filler-matrix interfacial adhesion, which can result in poor filler dispersion and large particle agglomeration, both being detrimental for the mechanical performance of the polymer composites. Furthermore, the presence of the hydrophilic fillers frequently promotes undesirable water uptake phenomena that can cause dimensional changes and ageing in the polymer composites [32]. As a result, different strategies have been used to enhance interfacial adhesion in polymer composites, improving their final properties [33]. Among them, filler surface modification with silane coupling agents [34,35,36], chemical treatments such as alkenyl succinic anhydride (ASA) or sodium hydroxide (alkaline treatment) [37,38], and plasma treatment [39], have been explored. Furthermore, promising results have been recently achieved by means of the addition of reactive compatibilizers during the extrusion or compounding process, the so-called reactive extrusion [40,41], which is also more cost-effective and easier to scale up. Among the different compatibilizers, the use of graft copolymers containing reactive groups, such as maleic anhydride (MAH) or glycidyl methacrylate (GMA), represents an effective way to improve the interfacial interaction between the polymer and filler and immiscible biopolymer blends [42]. For instance, based on these reactive compatibilizers, a feasible and robust strategy has been recently developed to prepare stable co-continuous blends of PBAT and PLA [43,44].

The use of the same polymer or a highly soluble one as the MAH-grafted matrix makes these agents very compatible with the polymer matrix, whereas the reactive groups can interact with the –OH groups present on the lignocellulosic filler surface, forming covalent bonds during manufacturing by melt mixing at high temperature. Using this approach, Tserki et al. [45] prepared and, thereafter, incorporated poly(butylene succinate-*co*-butylene adipate) grafted with maleic anhydride (PBSA-*g*-MAH) at 5 wt.% into green composites of PBSA with spruce and olive husk flours at 30 wt.%. Results demonstrated that an increase in the mechanical strength in the green composite ∼33% to 56% was observed. More specifically, for spruce flour composites, the tensile strength increased from 18.5 to 26.5 MPa, reaching values in the range of unfilled PBSA. In another study, Kennouche et al. [46] showed that a high compatibilization was achieved in composites of PHBV/PBS/halloysite nanotubes (HNTs) by poly(3-hydroxybutyrate-*co*-3-hydroxyvalerate) grafted with maleic anhydride (PHBV-*g*-MAH). Authors reported an improved dispersion of PBS nodules in the immiscible biopolymer blend with a preferential location of HNTs, owing to the diffusion and emulsifying effect of the PHBV-*g*-MAH chains. More recently, Muthuraj [47] described composites of PBS/PBAT with miscanthus fibers at 50 wt.% successfully compatibilized by previously prepared MAH-grafted blends of PBS/PBAT. In particular, tensile strength was increased by approximately 69% when compared to the uncompatibilized composite.

In the present study, green composites of PBS/PSF at different weight ratios, that is, 5, 10, 20, and 30 wt.% of filler, were prepared for the first time. In order to enhance the compatibility between PBS and PSF, poly(butylene succinate) grafted with maleic anhydride (PBS-*g*-MAH) was previously synthesized by a melt-grafting process induced by the presence of an organic peroxide as a radical initiator and, thereafter, added as reactive compatibilizer during melt extrusion at a constant 1/6 (*wt*/*wt*) ratio in relation to the filler content in the green composite. The resultant compounded pellets were finally shaped into pieces by injection molding and characterized to evaluate the performance of the newly developed green composites.

## 2. Results

### 2.1. Morphology of PSF

Particle morphology, that is, shape, size, and aspect ratio, plays an important role in the reinforcement of polymer composites. Figure 1 shows the morphology of the PSF particles observed by FESEM and their respective length and diameter histograms. One can observe in Figure 1a that the lignocellulosic particles displayed a rod-like morphology, although some platelets and ellipsoidal-like particles were also formed. A similar sheet-like structure was also observed by Bordbar and Mortazavimanesh [42]. The particles also showed a rough surface, which can be attributed to the grinding process performed on hard shells due to their high lignin composition. The particle surfaces also displayed high porosity, as it can be inferred from Figure 1b, showing a FESEM micrograph of the lignocellulosic particles taken at higher magnification. This morphology could be beneficial to interactions with the biopolymer matrix by acting as anchoring points. In this regard, da Silva et al. [48] reported a similar microstructure for the pistachio shell, indicating that both their surface and cross-section show a dense surface with few cracks. This observation is also in agreement with Yang and Lua [49], who showed that the surface of pistachio shell was quite dense without any pores except for some occasional cracks. Regarding the dimensions of the micronized particles, Figure 1c,d gather the histograms in terms of their length and diameter values, respectively. The particles presented an average value of length of 85.8 ± 47.2 µm, whereas the average diameter was 16.1 ± 8.8 µm, resulting in aspect ratio or length/diameter (L/D) of approximately 5. In both cases, the particle dimensions were defined by a classical monomodal distribution with a long tale for the fraction of larger particles. Therefore, this particle morphology is based on a relatively low size, which can offer a positive effect on the overall properties of the polymer composite. In this regard, Kadhim et al. [50] proposed that small lignocellulosic particles, of approximately 53 µm, promoted higher mechanical performance in comparison with particles larger than 100 µm in poly(methyl methacrylate) (PMMA) composites.

### 2.2. Optical Properies of PBS/PSF Composites

Figure 2 shows the resultant injection-molded pieces of the PBS and PBS/PSF composites at different filler loadings. This visual appearance is essential in the role that an end product plays in consumers and applications. One can observe that the unfilled PBS sample showed a “cream” color with no transparency. Then, the pieces developed a light brown color when the micronized powder of PSF was added to PBS, which was progressively intensified as a function of the filler content. This color variation can be mainly ascribed to the intrinsic yellow-to-brown color of PSF. However, the strong color variations, particularly at the highest filler contents, can also be influenced by the local high temperature increases that could occur during melt mixing of these formulations due to their higher specific mechanical energy.

Table 1 shows the results of the colorimetric measurements in terms of the L*a*b* color coordinates and the color difference, that is, Δ*E***_ab_*, between the PBS piece and the green composite pieces. One can observe that PBS exhibited a white-to-yellow tonality with values of L* = 83.8 and values of a* and b* of −1.8 (green) and 1.1 (yellow), respectively. The incorporation of 5 wt.% PSF resulted in a significant color change with a lower luminosity (L* = 71.8), which represented a percentage difference of 14.3%, and a brownish tonality with values of a* and b* of 1.1 (red) and 15.8 (yellow), respectively. The highest addition of PSF both significantly reduced L*, down to 57.8, and increased a* and b* up to values of 6.8 and 25.1, respectively. Therefore, the total color variation between PBS and the different green composite samples increased significantly with the filler content, and it was higher than 19 in all cases. According to the proposed visual evaluation [51], a color variation of $\Delta E_{ab}^{*}$ > 5 implied that an observer can easily notice different colors. Furthermore, the green composite pieces showed a similar color tonality than some natural woods, a characteristic key property of wood plastic composites (WPCs). For instance, oak without treatment shows color coordinates of L* = 64.46, a* = 6.77, and b* = 20.17, while pine treated at 160 °C presents coordinates of L* = 64.98, a* = 9.34, and b* = 27.14 [52].

### 2.3. Mechanical Properties of PBS/PSF Composites

Table 2 shows the mechanical properties of the PBS/PSF composite pieces obtained from the tensile tests. These results are usually of great interest to evaluate the effect of the lignocellulosic filler on the biopolymer and the performance of the composite pieces for different applications. The tensile test results were expressed in terms of tensile modulus (E), maximum tensile strength (σ_max_), and elongation at break (ε_b_).

As a reference point, the properties of the unfilled PBS pieces were analyzed, showing a ductile behavior with E and σ_max_ values of 598.8 and 27.8 MPa, respectively, being the ε_b_ value 225.5%. These values are similar to those reported earlier for injection-molded PBS pieces [53], whereas the differences can be ascribed to the molecular weight (M_W_) and crystallinity of the PBS sample. In this regard, Jin et al. [54] showed values of E, tensile strength at yield (σ_y_), and ε_b_ in the ranges of 280–400 MPa, 24–35 MPa, and 250–640%, respectively, with significant variations depending on the biopolymer’s M_W_. In particular, the mechanical properties increased with the number-average-molecular weight (M_n_) of PBS up to 4.1 × 10^4^ g/mol, which is similar to the grade used herein, and then decreased with further increasing of M_n_. Authors concluded indicating that the best mechanical performance is neither attained for the sample with the highest crystallinity nor for the sample with the highest M_W_. One can further observe that the incorporation of PSF increased the E value of the PBS pieces; however, it also resulted in a material with a lower deformation capacity so that the ductile properties of the injection-molded pieces were reduced. In this way, E values of 655.5 and 1039.6 MPa were obtained for the 10PSF and 30PSF pieces, which represents percentage improvements of approximately 9.5% and 79.6%, respectively, compared to the unfilled PBS piece. However, the effect of PSF on the tensile strength also followed a decreasing trend as a function of the amount of lignocellulosic filler. This variation resulted in σ_max_ values of 17.0 MPa for the 20PSF piece and 16.0 MPa for the 30PSF piece, which means percentage reductions of approximately 38.8% and 42.2%, respectively. In this regard, it can be considered that the presence of lignocellulosic particles inside the biopolymer matrix acted as a stress concentration and resulted in a lower tensile strength [50,55]. Furthermore, the ductility of the pieces was significantly reduced with the PSF content. In the case of the PBS-5PSF, the ε_b_ value decreased to 42.9%, being approximately 5 times lower than the neat PBS piece. The incorporation of higher amounts of filler further reduced the ductility of the material, but the percentage differences were smaller. For instance, the ε_b_ value for the PBS-30PSF piece, with 30 wt.% PSF, reached a value of 9.7%. This embrittlement phenomenon is commonly observed in polymer composites due to the reduction of the movement capacity of the polymer chains by the filler presence and it is particularly critical in the plastic deformation zone, which is drastically reduced [56,57]. The mechanical results reported herein agrees with previous works based on polymer composites filled with pistachio shells. For instance, the work of Kadhim et al. [50] reported that powder of pistachio shell could successfully reinforce PMMA, increasing E from nearly 1 GPa to up to 1.35 GPa, whereas σ_y_ and ε_b_ decreased from 66 MPa and 5% to approximately 46 MPa and 1.8%.

Table 2 also shows the results of the Shore D hardness tests. In the case of the unfilled PBS pieces, it showed a hardness value of 62.9. Similar to the E values, the incorporation of PSF to PBS yielded a progressive increase in the hardness of the pieces, which is related to the reduction of the movement of the biopolymer chains [58]. In particular, hardness increased from 63.6, for the PBS-5PSF piece, to 69.5, for that of PBS-30PSF. This Shore D hardness value is relatively close to, for instance, that of PLA, that is, ~79, which is considered to be a hard biopolymer material [51]. Finally, results of the Charpy impact test showed a significant reduction in the energy absorption capacity when the micronized lignocellulosic particles were introduced. For instance, whereas the neat PBS piece yielded an impact energy absorption of 11.4 kJ/m^2^, the addition of 5 wt.% PSF resulted in a value of 6.8 kJ/m^2^ while the use of 30 wt.% PSF resulted in a value of 2.9 kJ/m^2^, which respectively represents percentage reductions of approximately 40.4% and 74.6%. Impact strength is highly related to the deformation capacity of the material, and those materials with a higher deformation ability show a higher energy absorption in Charpy impact test. In comparison with a previous work in which PBS was filled with 30 wt.% of almond shell flour (ASF), the incorporation of the lignocellulosic particles also decreased impact strength from 16.5 to 1.8 kJ/m^2^; however, the use of maleinized linseed oil (MLO) as compatibilizer increased toughness in the green composite to 3.8 kJ/m^2^ [53].

### 2.4. Morphology of PBS/PSF Composites

In Figure 3, field emission scanning electron microscopy (FESEM) micrographs with the surface fractures of the green composite pieces after the Charpy impact tests can be observed. It can be seen that the morphology of the surface fracture changed from a rough surface, as seen in Figure 3a in the case of the unfilled PBS piece, to a smoother fracture surface with the presence of microcracks in the green composite pieces shown in Figure 3b–e. The latter morphology, with a lower roughness, can be ascribed to the embrittlement of the PBS matrix due to the presence of the lignocellulosic fillers [59]. It should also be underlined the absence of gap surrounding the PSF surface in contact with PBS since the lignocellulosic fillers were completely embedded in the biopolymer matrix, suggesting that PBS-*g*-MAH successfully compatibilized the two components of the green composite. Similar morphological observations have been previously reported for PBS/ASF compatibilized by MLO, who indicated that a good compatibilization is achieved when the size of these gaps are reduced or even removed [60]. Similar findings were also reported by Phua et al. [61], who similarly proposed PBS-*g*-MAH for improving the interaction of PBS in nanocomposites prepared with organically modified montmorillonite (OMMT).

### 2.5. Spectroscopic Properties of PBS/PSF Composites

Figure 4 shows the attenuated total reflection-Fourier transform infrared (ATR-FTIR) spectra of the PSF powder, the PBS-*g*-MAH dough, and the injection-molded pieces of neat PBS and its green composites with PSF and PBS-*g*-MAH. It can be seen in Figure 4a that the FTIR spectrum of PSF was mainly characterized by the presence of three strong peaks. It can be observed the first broad and intense band at ~1035 cm^−1^ wavelength, which can be correlated with C–O stretching vibrations in alcohols and phenols of lignocellulose, and it belongs to the polyglykosidic moieties (polyol) [48]. Then, the second strong peak arose at 2910 cm^−1^, and it corresponds to ν(C–H) vibrations in methyl (–CH_3_) and methylene (–CH_2_–) groups of cellulose and lignin [62]. This band has been indicated to be contrary to the δ(C–H) vibrational bands for –CH_3_ and –CH_2_– groups due to the skeletal C–C vibrations in aromatic rings, which were also found for pistachio nut shells and located herein at around 1263 and 1412 cm^−1^ [49]. The latter bands overlapped with those attributed to C–O–H in-plane bending and C–O stretching for carboxyl (–COOH) groups in lignin [48]. Finally, the third and also broad peak seen in the region of 3000–3600 cm^−1^ is indicative of the large presence of hydroxyl (–OH) groups in pistachio shell [63]. Other minor peaks were seen at 1664 cm^−1^, which occurs due to C=C vibrations in aromatic rings and olefinic ν(C–C) absorptions present in lignin, and at 1735 cm^−1^ due to the C=O stretching of the carbonyl (C=O) group, which is mainly associated with hemicellulose [48]. Therefore, from the FTIR spectrum of PSF, it was confirmed the lignocellulosic nature of the food waste derived filler as well as the presence of several oxygen groups on its chemical structure such as carbonyl groups, ethers, esters, alcohols, and phenol groups.

The characteristic peaks for PBS, in both the unfilled piece and the green composite pieces, were seen as a strong and sharp band at ~1714 cm^−1^ wavelength, ascribed to the C=O stretching of the ester groups in biopolyester, in addition to multiple peaks in the 1100–1200 cm^−1^ region mainly corresponding to the C–O stretching vibration in the ester groups, and the broad and low-intense peaks at 1330 and 2960 cm^−1^ due the symmetric and asymmetric stretching vibrations of –CH_2_– groups [54]. In relation to the strongest peak, it was reported that the PBS biopolymer can show three absorption bands at 1736, 1720, and 1714 cm^−1^ wavelengths that are stemmed from C=O stretching modes in the mobile amorphous fraction (MAF), rigid amorphous fraction (RAF) (or intermediate phase), and crystalline phase, respectively [64]. Since the crystal lattices should have the strongest confinement effect on C=O groups in the lamellar structure of the PBS crystal, including physical and hydrogen bonding, it was suggested that it is more reasonable to assign the 1714 cm^−1^ band to the C=O in crystal lattices. Therefore, this further confirms the high crystallinity achieved in the PBS samples during injection molding.

The outcome of the MAH grafting onto PBS was also investigated by FTIR. The strongest absorption signal was the one attributed to the C=O stretching vibrations of ester groups in PBS, which slightly shifted to ~1720 cm^−1^ and also narrowed due to crystalline phase disruption and potential loss. By comparison of the PBS-*g*-MAH and PBS spectra in the 1900–1500 cm^−1^ region, one can observe in Figure 4b that there was a new signal absorption band at 1849 cm^−1^, which is characteristic for succinic anhydride groups and has been formerly attributed to the symmetric C=O stretching bonds of MAH [65]. Furthermore, this was accompanied by a signal level increase at nearly 1780 cm^−1^, corresponding to their symmetric stretching. Moreover, the peak related to ν(C–H) vibration of PBS was shifted in the PBS-*g*-MAH spectrum from approximately 2965 cm^−1^ to 2980 cm^−1^ as a result of the grafting process since it could be affected by the =CH_2_ vibration of the cyclic MAH that was reported to arise at 3058 cm^−1^. Although the band changes in the PBS-*g*-MAH were subtle, these have already served to demonstrate the reactive grafting process of MAH at the diol unit of PBS [45,61]. All these newly formed bands and band distortions are, thus, representative of the functional groups available in the grafted biopolymer [66,67,68,69]. In addition to this, these new peaks did not appear in either the neat PBS spectrum or green composite spectra, suggesting that these groups were removed or consumed during melt grafting.

Finally, it can be observed that all the green composites showed very similar FTIR spectra than neat PBS, with the most noticeable change in the intensity increase of the peak centered at ~1163 cm^−1^. This band change can be ascribed to the presence of PSF in PBS, with a strong contribution of the C–O stretching vibration signals, which was seen at a lower wavelength in the spectrum of the biopolyester. Furthermore, the subsequent broadening of signals in the region of 1300–1400 cm^−1^, from which multiple peaks with low intensity arise, could be attributed to esters (e.g., R–CO–O–R′), ethers (e.g., R–O–R′), or phenol groups newly formed due to the grafting reaction of –OH groups of cellulose and lignin with the MAH groups present in PBS-*g*-MAH. Moreover, the intensity of the peaks observed in the 1150–1300 cm^−1^ decreased, which has been ascribed to −C–O–C− groups in the ester linkages of PBS [61], suggesting that the proportional contribution of the original groups in the biopolyester was reduced and also observed at higher wavenumbers due to the potential formation of new ones. Therefore, these peaks variations point to an esterification reaction achieved by the functional MAH groups present in PBS-*g*-MAH with the −OH groups of PSF. The scheme of this process is proposed in Figure 5, which illustrates the grafting mechanism of PSF onto PBS by PBS-*g*-MAH. Although PBS-*g*-MAH is readily soluble in PBS, part of their multiple MAH can also react with the −OH end groups of the biopolyester, both present in acids or alcohols, following the scheme shown in Figure 5a. Moreover, according to the final scheme proposed in Figure 5b, other MAH groups can connect with –OH groups present in the cellulose or lignin of PSF by esterification [70]. Therefore, the here-developed PBS-*g*-MAH acted as a bridge between the PSF filler and the PBS matrix.

### 2.6. Thermal Properties of PBS/PSF Composites

Figure 6 shows the differential scanning calorimetry (DSC) curves for the injection-molded pieces of PBS and its green composites with PSF in the thermal range from 140 °C to 20 °C during cooling (Figure 6a) and from 40 °C to 175 °C in the second heating (Figure 6b). In Table 3, the main thermal parameters obtained from the cooling and heating thermograms are summarized, that is, the crystallization temperature (T_C_), cold crystallization temperature (T_CC_), melting temperature (T_m_), and the amount of crystallinity (X_C_) determined from the melting enthalpy (ΔH_m_) of the samples. One can observe that the neat PBS sample first crystallized from the melt at 69.7 °C and, thereafter, further cold crystallized at 96.6 °C prior to melting at 113.4 °C. This melting peak value is in the range established for the semi-crystalline PBS homopolymer, showing values from 112 to 116 °C [7] and also in agreement with those obtained by Kim et al. [71] and Chen et al. [72] who reported values of 112.2 °C and 114.2 °C, respectively. The formation of PBS crystals during both cooling and also, to a lesser extent, heating, the so-called cold crystallization, has been reported elsewhere, showing T_C_, T_CC_, and T_m_ values of 80.5, 102.5, and 114.6 °C, respectively [73]. In particular, cold crystallization is a phenomenon mainly associated with high-M_W_ PBS due to the lower number of chain ends involved and, subsequently, reduction in the free volume [74,75]. The incorporation of PSF promoted crystallization from the melt but also slightly delayed cold crystallization, particularly at low filler contents, reaching the highest T_C_ and T_CC_ values at 77.7 and 102 °C, respectively, for the PBS-5PSF sample. This observation suggests that the presence of the lignocellulosic fillers favored crystallization of the PBS molecules. However, the values of T_m_ were significantly unaltered whereas the percentage of crystallinity, measured as X_C_, increased from 51.8%, for the neat PBS, up to nearly 63% for the green composites containing 20 and 30 wt.% PSF. This result indicates that the lignocellulosic fillers acted as heterogeneous nuclei during the formation of crystals and a greater number of molecules could crystallize from the melt. The nucleating effect on PBS of other fillers, such as calcium carbonate and distiller grains, has been previously reported by Chen et al. [72] and other authors [76,77]. Similar results were also observed by Liminana et al. [78], who reported an increase in the PBS crystallinity after the incorporation of ASF in different contents from 10 to 50 wt.%. Furthermore, the filler effect on crystal growth could also be maximized by the use of the PBS-*g*-MAH compatibilizer that, as shown above during the morphological analysis, successfully enhanced the interactions between the PBS matrix and the lignocellulosic particles [79]. In this regard, Phua et al. [61] suggested that the improved interaction between PBS and fillers by PBS-*g*-MAH contributed to the crystallization increase of the biopolyester.

Figure 7 shows the thermogravimetric analysis (TGA) and first derivative thermogravimetry (DTG) curves, respectively gathered in Figure 7a,b, for the PBS pieces and their green composite pieces with different contents of PSF. Table 4 presents the main thermal parameters obtained from the thermogravimetric curves, that is, the onset degradation temperature that was considered for the temperature with a loss of 5% (T_5%_), temperature of maximum degradation (T_deg_), corresponding to the maximum degradation rate and identified in the corresponding DTG curves, and residual mass at 700 °C. In the case of PSF, like other biomass, three different mass losses were seen during the degradation process, which have been ascribed to drying, devolatilization, and charring [80]. The first mass loss, at ~100 °C, is related to the removal of moisture present in the lignocellulosic material. The second one, in the thermal range from 200 to 400 °C, represents the thermal decomposition of holocellulose, that is, the degradation of hemicellulose (210–325 °C) and cellulose (310–400 °C) [81]. This zone is known as the “active pyrolysis zone” due to its high rate of devolatilization, where occurs the majority of the pyrolysis with a mass loss of ~55% [40]. Finally, the third zone of degradation started at approximately 310 °C, and it corresponds to the thermal decomposition of the rest of cellulose and lignin, which occurs from 400 °C. This stage is termed “passive pyrolysis” since the mass loss rate is lower compared with the previous stage, and it can be seen as a tailing section in the DGT curve [81].

One can observe that PBS was thermally stable up to ~297 °C and, opposite to other reports [61,82], thermal degradation was seen to occur in two steps. Nevertheless, a two-step thermal decomposition behavior has been widely observed in biopolyesters [83,84]. It can be seen that the main and fast mass loss took place from nearly 300 to 405 °C, and it corresponds to the main degradation of the biopolymer backbone, yielding the formation of char with a mass loss of over 90%. The second decomposition step was observed in the 405–525 °C range, and it is related to the thermo-oxidative degradation of the char produced during the first degradation step, associated with a loss of nearly the totality of the remaining biopolyester. This degradation profile has been reported to occur in thermostable polyesters subjected to air atmosphere [85]. Furthermore, the thermal stability of the green composites, interestingly, revealed that the samples with low filler loadings, that is, filled with 5 and 10 wt.% PSF, significantly delayed T_5%_. In particular, the PBS-5PSF and PBS-10PSF green composite pieces yielded values of 341.9 °C and 318.3 °C. Meanwhile, the PBS samples filled with 20 and 30 wt.% PSF, that is, PBS-20PSF and PBS-30PSF, achieved slightly, but still significant, lower onset values of degradation than the neat PBS sample (290.9 °C and 279.6 °C, respectively). This result suggests that, at low filler contents, the good filler-to-matrix interaction achieved by PBS-*g*-MAH induced a positive effect on thermal degradation due to grafting of the PBS molecules onto the lignocellulosic particles. However, the highest content of fillers slightly impaired the thermal degradation of PBS, which is an effect reported and ascribed to the inherently lower thermal stability of lignocellulose [86]. It is also worth mentioning that, in all cases, due to the presence of the lignocellulosic fillers, the char mass produced during the second degradation step increased. Therefore, in terms of the residual mass, measured at 700 °C, it increased as a function of the PSF loading incorporated in the samples, from 0.4 wt.% for neat PBS, up to a value of 6.6 wt.% in the PBS-30PSF piece. This residual mass corresponds to inorganic residues, such as silica, contained in PSF [72].

### 2.7. Thermomechanical Properties of PBS/PSF Composites

Figure 8 shows the dynamical mechanical thermal analysis (DMTA) curves for the neat PBS pieces and the PBS/PSF composite pieces. Table 5 gathers the most relevant thermomechanical parameters, namely the storage modulus measured at −45, 25, and 70 °C, which correspond to temperatures below and above each transition region, and also the dynamic damping factor (*tan δ*). In Figure 8a, which shows the evolution of the loss modulus as function of temperature, it can be observed that the PBS piece presented at −45 °C the highest value, that is, 1973.3 MPa. This high value of storage modulus is due to the amorphous region of the biopolyester at this temperature being in its vitreous state. At higher temperatures, the PBS and its green composites showed two relaxation regions at approximately −40 °C and 30 °C, and these low- and high-temperature transitions are respectively related to its β- and α-relaxations. In particular, the low-temperature transition has been ascribed to the β-relaxation of the amorphous fractions of PBS, and it is also considered the glass transition of the biopolyester, for which a T_g_ value of −30 °C has been reported elsewhere [87]. Therefore, in the temperature range from −40 to 25 °C, the storage modulus suffered the most pronounced drop, reaching a value of 440 MPa. Thereafter, from nearly 35 °C, a low-intense reduction in the storage modulus was observed, which is indicative of the α-relaxation process related to the PBS polymer crystalline fractions [47]. Thus, the storage modulus of the PBS pieces dropped to a value of 214.3 MPa at 70 °C, which is close to the initiation of cold crystallization reported above during DSC analysis. One can further see that the incorporation of PSF reinforced the PBS matrix, and this effect can be easily noticed by the increase in the storage modulus along the whole thermal range tested. Thus, the highest value was attained for the PBS with 30 wt.% of PSF, showing a value of 2161.1 MPa at −45 °C. This observation agrees with the previous mechanical results, indicating that the lignocellulosic fillers reinforced the PBS matrix and restricted the movement of the biopolymer chains [78]. This result also points out that the use of the PBS-*g*-MAH as compatibilizer facilitates the interaction of PSF with PBS.

Figure 8b provides the evolution with temperature of *tan δ*, also called damping factor, which allows to identify the main relaxations of PBS. As it can also be observed in the table, the neat PBS showed a *tan δ* peak of −23.3 °C, which is related to its T_g_ and ascribed to the β-relaxation, whereas it was followed by a low-intense second peak at nearly 45 °C due to the α-relaxation. It can also be observed that the position of the damping factor peaks slightly increased to higher temperatures with the incorporation of PSF, which may indicate that these fillers caused some inhibition of the PBS chain motion [88]. As suggested above, this effect can also be ascribed to the higher interaction achieved due to the reactive grafting with PBS-*g*-MAH. A similar behavior was reported by Liminana et al. [60] with an increase of T_g_ in samples of PBS reinforced with ASF and compatibilized with MLO. Furthermore, it is worth mentioning that all the green composite samples presented lower values of *tan δ* peak than the unfilled PBS sample, decreasing progressively with the PSF content. This reduction is characteristic to materials with lower energy dissipation and reduced toughness, which can be explained by the fact that the biopolymer with amorphous regions was partially replaced with hard lignocellulosic fillers [89].

### 2.8. Water Uptake of PBS/PSF Composites

Figure 9 shows the evolution of water uptake with time in a 14-week immersion period for the PBS pieces and the PBS/PSF composite pieces. In the case of the neat PBS piece, it reached an absorption value of ~0.73 wt.% after 14 weeks of immersion in water. A similar value was reported previously by Frollini et al. [90], confirming the high hydrophobicity of this biopolyester. Furthermore, it can be observed in the graph that the introduction of the lignocellulosic fillers notably increased the amount of water absorbed in the material during the immersion period. In general, a direct relationship can be observed between the amount of this type of waste derived filler, based on lignocellulose and the total water absorbed in the polymer composite sample [91]. This fact is ascribed to the hydrophilic nature of lignocellulose due to the large amount of free –OH groups on the filler surface [92]. For instance, water uptake increased up to 2.23 wt.% for the green composite pieces filled with 10 wt.% PSF, that is, PBS-10PSF, whereas it reached a value of 6.14 wt.% for a 30 wt.% filler loading, that is, PBS-30PSF. In this regard, Gairola et al. [93] also reported that PSF shows a great tendency to entrap water, showing a value of water absorption of 2.75 wt.% after 7 days for a thermosetting epoxy composite with 10 wt.% PSF. Nevertheless, it is worth remarking that all these tests were carried out using injection-molded pieces that were laterally uncoated when fully immersed in water, which may represent more aggressive conditions than those typically found for end-use applications of WPCs.

One can also observe that, at the end of the immersion period analyzed, that is, 14 weeks, all samples saturated their water absorption capability. However, the largest mass increases occurred in the first weeks of immersion and, interestingly, this was seen to occur more rapidly in the unfilled PBS sample and the green composites with lower filler loadings. For example, the green composite pieces containing 5 wt.% PSF saturated after approximately 5 weeks, whereas water uptake was stabilized after 9 weeks of immersion in the green composite pieces filled with 20 and 30 wt.% of lignocellulosic fillers. In general terms, the use of reactive compatibilizers that show the capacity of anchoring the lignocellulosic fillers to the polymer matrix, which is the case of PBS-*g*-MAH, can slightly reduce water uptake due to the reduction in the number of freely available –OH groups [94]. However, better results in terms of hydrophobicity have been achieved by chemical pre-treatments on the filler surface [95]. For instance, it has been reported that lignocellulosic waste flour pre-treated with acetylation or propionylation led to a significant reduction in the percentage of water uptake in PBS green composites, of nearly 50%, whereas the use of compatibilizers yielded lower reductions. This fact was attributed to the substitution of hydrophilic –OH groups of the lignocellulosic material with acetyl and propionyl groups, rendering the lignocellulosic flour surface more hydrophobic [96]. In the case of the compatibilizer addition, the water absorption reduction can be mainly attributed to the formation of covalent bonds between the MAH groups and the –OH groups at flour surface [97], which are therefore less effective than chemically modified lignocelluloses.

### 2.9. Disintegration in Controlled Compost Soil of PBS/PSF Composites

The disintegration behavior of the PBS and PBS/PSF pieces under controlled composting conditions was analyzed for a period of 112 days, that is, 16 weeks. This timeframe goes beyond the “reasonably short period of time” described in international standards of compostability. In particular, the EN 13432 or ASTM D6400 standards require the plastic articles certified as compostable to disintegrate after 12 weeks or 84 days, and completely biodegrade after 180 days, which means that at least 90% of the bioplastic material will have been converted to carbon dioxide (CO_2_), water, and biomass. Figure 10 shows results of the disintegration tests in terms of the evolution of the percentage of weight loss (Figure 10a) and the visual aspect of the pieces at each tested time during composting (Figure 10b). It can be observed that, during the first 4 weeks of composting, the mass loss was very subtle, and the visual aspect of the pieces remained nearly unaltered. Then, higher rates of disintegration were observed after nearly 35 days of composting, from which all the PBS samples started to loss mass relatively fast whereas the piece surfaces were seen to be eroded. From the week 6 up to approximately 3 months, the pieces showed a linear and very steep degradation progression. At this time, however, the degradation rate was noticeably reduced, and the mass loss reached a value of approximately 20 wt.% at the end of the test. In this regard, it is worth noting that the test was conducted at 58 °C and 55% RH, which are certainly favorable conditions in terms of high temperature and humidity for the hydrolytic degradation of PBS. On the base of the here-attained uncompleted disintegration process, other authors [98,99] also found a similar degradation profile for PBS, identifying three phases with different degradation speeds, namely the lag phase with a low rate (<5 days), biodegradation phase where the rate was accelerated (6–66 days), and *plateau* phase in which disintegration leveled off (>67 days). Authors also concluded that the shape and format of the PBS articles played a major role during disintegration in controlled compost. In particular, whereas PBS powder with an average particle size of 42 μm and the resultant 40 μm thermo-compressed film sizing 1 cm x 1 cm showed biodegradation percentages after 90 days of 71.9 and 60.7 wt.%, respectively, the as-received granules of ~3 mm only disintegrated up to 14.1 wt.%. It was then confirmed that biodegradability of PBS under composting conditions is mainly controlled by the sample’s specific surface area. In another study, it was observed that PBS sheet samples of 30 mm × 30 mm × 1 mm were discolored after 1 month and their surface was completely damaged by the microorganisms of biodegradation after 3 months, but the sheets did not fully disintegrate in 5 months [45] Therefore, the previous finding supports and agrees with the percentages of degradation achieved herein for the injection-molded pieces of PBS with a thickness of 4 mm, which was approximately 13.4 wt.% for 91 days and 18.3 wt.% at the end of the test, that is, after 112 days.

One can also observe that the incorporation of PSF together with PBS-*g*-MAH increased the disintegration rate during the second biodegradation phase, that is, from days 35 to 70. Then, disintegration progressively increased with the filler content, and this effect may be ascribed to the higher tendency of the fillers to entrap water, as described above during the water uptake analysis, which could then transfer water into the composite by means of the capillary effect favoring the biopolymer hydrolysis of the ester groups [100]. Nevertheless, around day 60, it can be observed that this tendency changed, and the disintegration rate of PBS increased faster than in the case of the green composites. These mass loss changes were correlated with the visual images of the pieces at different times of disintegration, where it can be seen that the unfilled PBS fragmented to a higher extend at the end of the assay, although the development of color dark-to-black changes occurred earlier in the case of the green composites. A similar trend was observed by Quiles-Carrillo et al. [14] during the biodegradation process of green composites of PLA with orange peel flour (OPF) compatibilized by MLO, also showing a slightly slower degradation profile in the final weeks for the green composites in comparison to the unfilled biopolyester. The slower degradation rate in the green composites can be mainly ascribed to a latter increase in water diffusion resistance through the composite due to a tortuosity effect of the fillers on the biopolymer matrix during the *plateau* phase [101]. Furthermore, the use of PBS-*g*-MAH could reduce the –OH concentration and, thus, reduce the water activity. Other authors further suggested that the presence of remaining DCP amounts, which was used herein as radical initiator for the synthesis of PBS-*g*-MAH, could also interfere with the biopolymer chain scission, delaying the biodegradability process [102]. Additionally, it has also been indicated that the presence of diverse mineral fillers, such as talk and chalk, which can be present in minor quantities in pistachio shell, together with silica, can also influence the disintegration behavior of PBS [98,99]. In any case, one should conclude that the large thickness of the pieces certainly limited the appropriate disintegration evaluation of PBS under industrial composting conditions so that the actual effect of the lignocellulosic fillers on compostability was difficult to elucidate.

## 3. Discussion

The present study demonstrates the high potential of pistachio shells, a residue of the agricultural and food industries, to be manufactured in the form of flour for serving as a reinforcing filler in combination with biodegradable polymers to yield green composites in the frame of the Circular Bioeconomy. The incorporation into PBS by melt mixing of low and moderate contents of PSF, below 20 wt.%, and subsequent shaping using injection molding successfully led to the development of material pieces with a wood-like appearance, higher rigidity and hardness, improved crystallinity, and high thermal stability. To improve the inherently low compatibility between this waste derived lignocellulosic, PBS-*g*-MAH was synthetized using DCP as an initiator in a previous step. The grafting methodology achieved herein offers the possibility to produce tailor-made compatibilizers for different green composites since these can be produced with the same biopolymer as the base resin, so that it is fully miscible with the composite matrix, whereas it contains multiple reactive MAH groups that can chemically interact with the –OH groups in the cellulose or lignin present on the filler surface by esterification or with those of the biopolyester. As a result of this melt-grafting process, the interfacial filler-matrix adhesion is improved, and the properties of the green composites are enhanced and/or higher filler contents can be loaded. Future works will be dealing with the development of new green composites based on this approach, including the valorization of other agricultural, food, or marine wastes, and also a deeper understanding of their degradation mechanisms in different environments.

## 4. Materials and Methods

### 4.1. Materials

PBS Bionolle 1020MD was supplied by Showa Denko Europe (Munich, Germany), which is a liner aliphatic biopolyester characterized by a density of 1.26 g/cm^3^ and a melt flow rate (MFR) of 25 g/10 min (190 °C and 2.16 kg). It shows a M_n_ of 3.7 × 10^4^ g/mol and a dispersity (Ð = M_W_/M_n_) of 1.65 [103]. This PBS grade is mainly designed for injection molding and it is certified as compostable according to the EN-13432 standard.

PSF was obtained from Micronizados Vegetales S.L. (Córdoba, Spain) as a waste of the food industry. The pistachio shells were cleaned and ground into powder form using a jaw crusher. The collected powder was then sieved to obtain a flour composed of particles with a mean size of less than 75 µm. The raw pistachio shells and the resultant flour in powder form are shown in Figure 11.

MAH and DCP, with purity of 98%, were both purchased at Sigma-Aldrich S.A. (Madrid, Spain) in the form of fine powder.

### 4.2. Grafting Procedure

The grafting reaction was carried out an internal mini-mixer (HAAKETM PolyLabTM QC, Thermo Fisher Scientific, Karlsruhe, Germany) under the conditions established by Phua et al. [61] in the presence of an organic peroxide initiator. Initially, the PBS pellets were physically premixed with the MAH and DCP powders at contents of 10 and 1 parts per hundred resin (phr) of PBS, respectively. Then, the resultant mixture was fed into the melt-mixing device and processed at 135 °C for 7 min. Thereafter, the resultant dough was purified by refluxing in chloroform (Panreac S.A., Barcelona, Spain) for 4 h, and the hot solution was filtered and precipitated into cold methanol (Sigma-Aldrich S.A.). Finally, in order to remove any unreacted reagents, it was washed with methanol several times, followed by drying at 60 °C for 24 h in an air-circulating oven CARBOLITE Eurotherm 2416 CG (Hope Valley, UK).

The degree of grafting (G_d_) for PBS-*g*-MAH was determined through titration also following the determination proposed by Phua et al. [61]. Briefly, 1 g of purified PBS-*g*-MAH was refluxed for 1 h in 100 mL of chloroform. Then, 10 mL of distilled water were added and titrated immediately with 0.025 M potassium hydroxide (KOH, Sigma-Aldrich S.A.) using phenolphthalein (Fisher Scientific SL, Madrid, Spain) as indicator. G_d_ was calculated using Equation (1):
(1)$$G_{d}\left(\%\right)=\left[\frac{N\cdot (V_{1}-V_{0})\cdot 98.06}{1000\cdot W\cdot 2}\right]\times 100$$
where *N* is the KOH concentration [M], *V*_0_ and *V*_1_ represent the KOH volume [mL] for blank solution and for titration of PBS-*g*-MAH, respectively, and *W* is the sample weight [g]. A grafting efficiency or G_d_ value of 3.84 ± 0.27% was attained.

Figure 12 shows the reaction mechanism to obtain the grafted material, that is, PBS-*g*-MAH. The reaction starts with the peroxide decomposition to form the primary DCP free radicals (I) that abstract the hydrogen atom from PBS backbone to yield PBS macroradicals during the initiation step (II). These biopolymer macroradicals are formed mainly from chain transfer reactions by free radicals that are generated from thermal decomposition of organic initiators [104]. The resultant macroradicals propagated and set off the grafting of MAH onto PBS (III). The reaction continued until the resultant PBS-MAH macroradicals might undergo hydrogen transfer from another biopolymer chains, MAH, or the initiator and formed the so-called PBS-*g*-MAH (IV). Alternatively, the PBS-MAH macroradicals can react with other radicals in the system, such as MAH, PBS, or primary radicals to form a different structure of PBS-*g*-MAH (V). The formation of side groups and side chains is mainly due to addition of PBS macroradicals to the double bond of MAH molecules and also the combination of PBS macroradicals with those of PBS-MAH. In this regard, it should be considered that other unwanted reaction pathways could occur, for example, the homopolymerization of MAH, that is, PBS-(MAH-)_n_, resulting in a product with higher grafting degree but lesser compatibilization efficiency.

### 4.3. Preparation of PBS/PSF Composites

PBS, PBS-*g*-MAH, and PSF were first dried at 50 °C for 24 h in an air-circulating oven CARBOLITE Eurotherm 2416 CG to avoid hydrolysis by moisture during the extrusion process. Table 6 shows the compositions used for the compounding process, where PBS-*g*-MAH was added at a constant 1/6 (*wt*/*wt*) ratio in relation to the PSF content. The mixtures were weighed and pre-mixed manually in a zip bag that were fed into a co-rotating extruder of Mecánicas Dupra S.L. (Castalla, Spain). This extruder has a diameter of 25 mm and a L/D ratio of 24. Compounding was carried out at 25 rpm and temperatures of 115–120–125–130 °C (from the hopper to the die), resulting in a residence time of approximately 1 min.

Test specimens were thereafter obtained by injection molding using a Meteor 270/75 from Mateu & Solé (Barcelona, Spain) machine. The processing conditions consisted of a temperature profile of 120–125–130–135 °C (from the hopper to the injection nozzle), cavity filling and cooling times of 1 and 10 s, respectively, and a clamping force of 75 t. Standard samples with an average thickness of 4 mm were produced.

### 4.4. Characterization of PBS/PSF Composite Pieces

#### 4.4.1. Morphological Characterization

The morphology of the PSF particles and the fractured samples obtained from the Charpy tests were studied by field emission scanning electron microscopy (FESEM) in a ZEISS ULTRA 55 microscope from Oxford Instruments (Abingdon, UK) using an acceleration voltage of 2 kV. Prior to observation, the samples were sputtered with a gold-palladium alloy in an EMITECH sputter coating SC7620 model from Quorum Technologies, Ltd. (East Sussex, UK). The FESEM measurements were carried out using an acceleration voltage of 2 kV. Size values were obtained using the ImageJ program in 1.8 version from the National Institutes of Health (Bethesda, MD, USA) using, at least, 20 FESEM images in their original magnification.

#### 4.4.2. Color Characterization

Color changes were measured with a Konica CM-3600d Colorflex-DIFF2, from Hunter Associates Laboratory, Inc. (Reston, VA, USA). The CIELAB color space (L*, a*, b*) was determined in which L* refers to lightness of the sample while the coordinates a* stands for the green (a* < 0) to red (a* > 0) and b* for the blue (b* < 0) to yellow (b* > 0). Equation (2) shows the expression used to obtain the color differences compared with the neat PBS piece.
(2)$$\Delta E_{ab}^{*}=\sqrt{{\left(\Delta L^{*}\right)}^{2}+{\left(\Delta a^{*}\right)}^{2}+{\left(\Delta b^{*}\right)}^{2}}$$

Evaluation of color change was assessed following previous criteria [51]: unnoticeable (Δ*E***_ab_* < 1), only an experienced observer can notice the difference (Δ*E***_ab_* ≥ 1 and < 2), an unexperienced observer notices the difference (Δ*E***_ab_* ≥ 2 and < 3.5), clear noticeable difference (Δ*E***_ab_* ≥ 3.5 and < 5), and the observer notices different colors (Δ*E***_ab_* ≥ 5).

#### 4.4.3. Mechanical Characterization

The tensile properties of the PBS/PSF pieces with dimensions of 150 mm × 10 mm × 4 mm were obtained using the ELIB 50 universal test machine from Ibertest (Madrid, Spain) according to ISO 527-1:2012 with a 5-kN load cell and a testing speed of 5 mm/min. Hardness was measured according to ISO 868:2003 with a Shore-D scale using a 676-D hardness tester from J. Bot Instruments (Barcelona, Spain) on rectangular samples sizing 80 mm × 10 mm × 4 mm. Finally, impact strength was measured according to ISO 179-1:2010 with a 1-J pendulum from Metrotec S.A. (San Sebastian, Spain), using V-notched samples with a radius of 0.25 mm and sizes of 80 mm × 10 mm × 4 mm. All measurements were carried out at room temperature and, for each of the test, at least 6 measurements were performed.

#### 4.4.4. Infrared Spectroscopy

Chemical analysis of the PSF powder, PBS-*g*-MAH, and the pieces of PBS and PBS/PSF composites was performed by ATR-FTIR spectroscopy. For the test, a Bruker S.A Vector 22 (Madrid, Spain) was used, coupled to a PIKE MIRacle^TM^ single reflection diamond ATR accessory (Madison, WI, USA). Spectra were collected as the average of 10 scans between 4000 and 500 cm^−1^ with a resolution of 2 cm^−1^.

#### 4.4.5. Thermal Analysis

DSC was carried out in a Mettler-Toledo 821 calorimeter (Schwerzenbach, Switzerland). All the tests were performed with samples with an average weight of 6–7 mg, and the thermal program was divided into three stages: a first heating from 25 to 130 °C, followed by a cooling to 0 °C, and a second heating to 250 °C, all with all heating/cooling rates at 10 °C/min. Tests were run in nitrogen atmosphere with a flow-rate of 66 mL/min using standard sealed aluminum crucibles with a capacity of 40 μL. The degree of crystallinity ($X_{C}$) was measured following Equation (3):
(3)$$X_{C}=\left[\frac{\Delta H_{m}-\Delta H_{cc}}{\Delta H_{m}^{0}\times \left(1-w\right)}\right]\times 100$$

Where Δ*H_m_* (J/g) is the melting enthalpy, $\Delta H_{m}^{0}$ (J/g) represents the theoretical enthalpy of a fully crystalline sample of PBS with a value of 110.3 J/g [53,105], and the term 1 − *w* is the weight fraction of PBS.

Thermal degradation of the green composite pieces was assessed by TGA in a LINSEIS TGA 1000 (Selb, Germany). All samples, with a weight of 15–17 mg, were placed in 70 µL alumina crucibles, a dynamic heating program from 40 to 700 °C at a heating rate of 10 °C/min in air atmosphere. All thermal tests were performed in triplicate to obtain averaged results.

#### 4.4.6. Thermomechanical Characterization

Dynamical mechanical thermal analysis (DMTA) was carried out in a DMA1 dynamic analyzer from Mettler-Toledo (Schwerzenbach, Switzerland), working in single cantilever flexural mode. Rectangular samples sizing 20 mm × 6 mm × 2.7 mm were subjected to a dynamic temperature sweep from −45 to 75 °C and a heating rate of 2 °C/min. The selected frequency was set at 1 Hz, and the maximum deformation was 10 µm.

#### 4.4.7. Water Absorption Test

Injection-molded samples of 4 mm × 10 mm × 80 mm were immersed in distilled water at room temperature. Samples were first weighted in a balance and then immersed in distilled water. All of them were wrapped within a metal grid to ensure the correct immersion. Thereafter, the weight of all the sample pieces was measured in intervals of different time intervals for up to 14 weeks. For each measurement, prior to annotate weight, the surface moisture of the samples was removed with tissue paper. The amount of absorbed water during the process was calculated following Equation (4):
(4)$$\Delta m_{t}\;\left(\%\right)=\frac{W_{t}-W_{0}}{W_{0}}\times 100$$
where W_0_ and W_t_ represent the initial weight before the immersion and the weight of the sample after each immersion time.

#### 4.4.8. Disintegration Test

Disintegration of the PBS/PSF composite pieces was evaluated under composting conditions at 58 °C and 55% RH following the recommendations of ISO 20200. Injection-molded pieces sizing 20 mm × 20 mm × 4 mm were placed in a carrier bag and buried in a controlled soil compost made of sawdust (40 wt.%), rabbit-feed (30 wt.%), ripe compost (10 wt.%), corn starch (10 wt.%), saccharose (5 wt.%), corn seed oil (4 wt.%), and urea (1 wt.%). Once a week, each sample was unburied from the composting facility, washed with distilled water, dried, and weighed in an analytical balance. The weight loss during disintegration was calculated using Equation (5):
(5)$$Weight\;loss\;\left(\%\right)=\frac{W_{0}-W_{t}}{W_{0}}\times 100$$
where W_0_ is weight of the sample before the immersion and W_t_ is the weight of the sample in each measure. All tests were carried out by triplicate to obtain an average.

## Figures and Tables

**Figure 1 molecules-26-05927-f001:**
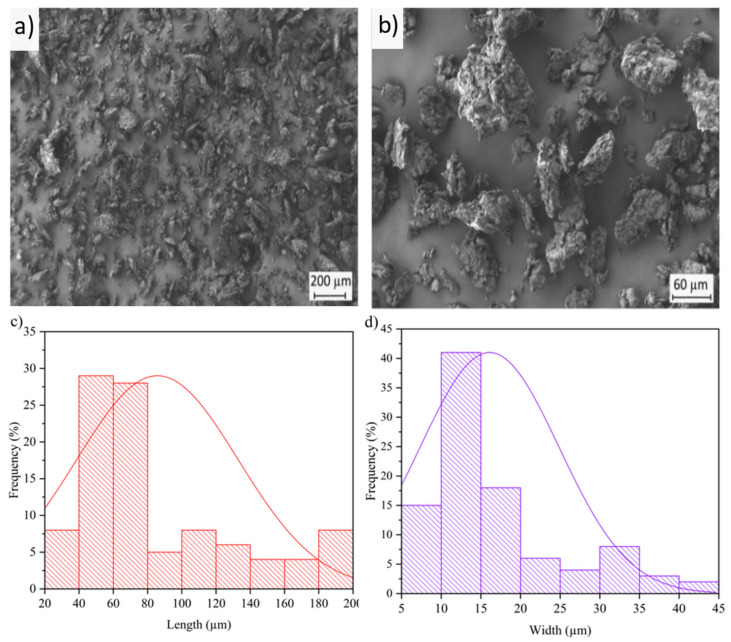
(**a**) Field emission scanning electron microscopy (FESEM) images of micronized pistachio shell flour (PSF) taken with a magnification of 50× and a scale marker of 200 µm; (**b**) FESEM image of PSF taken with a magnification of 200× and a scale marker of 60 µm; (**c)** histogram of the PSF particle length; (**d**) histogram of the PSF particle width.

**Figure 2 molecules-26-05927-f002:**
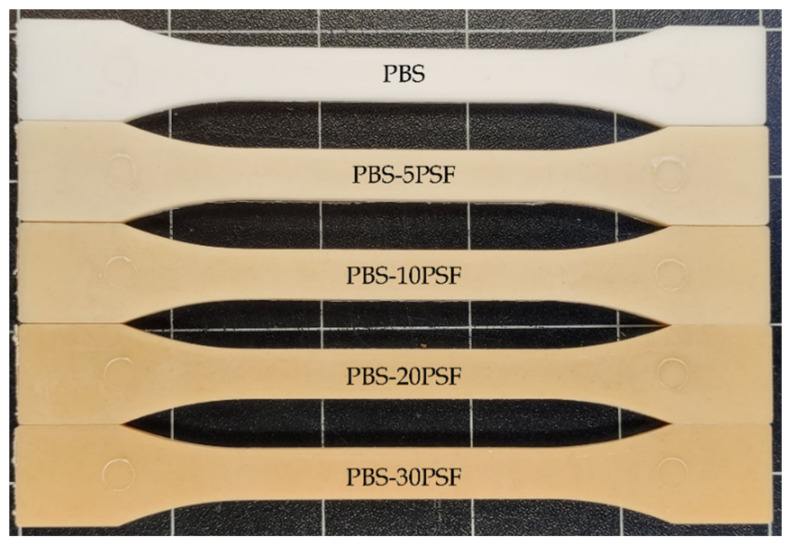
Visual appearance of the injection-molded pieces of poly(butylene succinate) (PBS)/pistachio shell flour (PSF) compatibilized with poly(butylene succinate) grafted with maleic anhydride (PBS-*g*-MAH).

**Figure 3 molecules-26-05927-f003:**
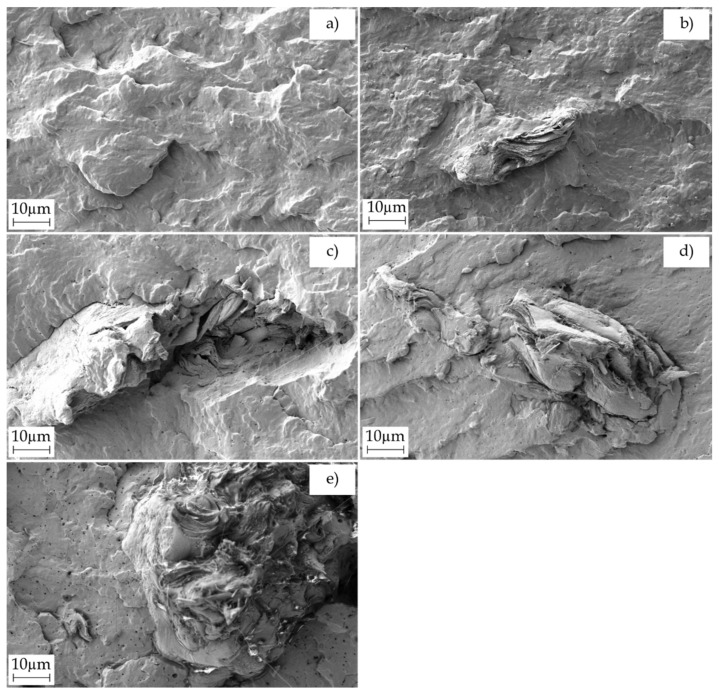
Field emission scanning electron microscopy (FESEM) images of the fracture surfaces of the injection-molded pieces of poly(butylene succinate) (PBS)/pistachio shell flour (PSF) compatibilized with poly(butylene succinate) grafted with maleic anhydride (PBS-*g*-MAH): (**a**) PBS, (**b**) PBS-5PSF, (**c**) PBS-10PSF, (**d**) PBS-20PSF, and (**e**) PBS-30PSF. Images were taken at 1000× with scale markers of 10 µm.

**Figure 4 molecules-26-05927-f004:**
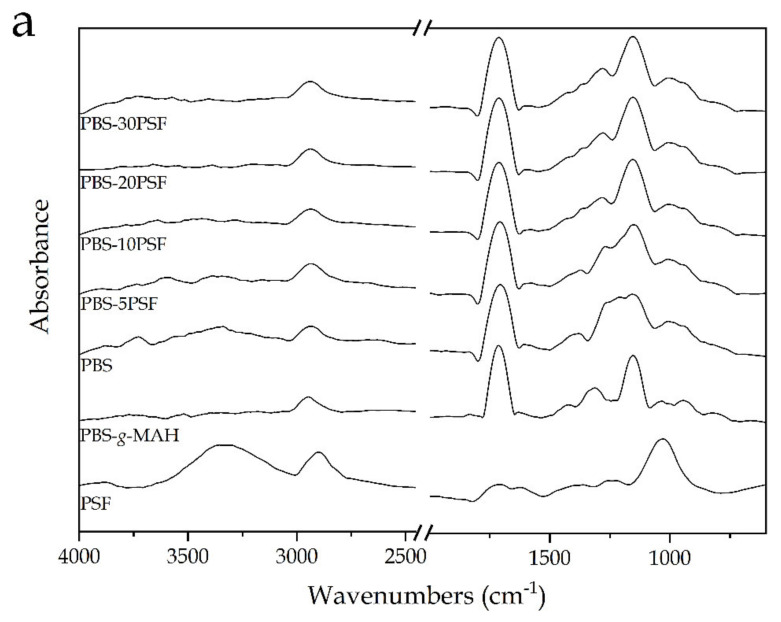
(**a**) Fourier transform infrared (FTIR) spectra, from bottom to top, of pistachio shell flour (PSF); poly(butylene succinate) grafted with maleic anhydride (PBS-g-MAH), poly(butylene succinate) (PBS), and PBS-5PSF, PBS-10PSF, PBS-20PSF, and PBS-30PSF green composites; (**b**) Detail of the PBS-*g*-MAH and PBS spectra in the 1900–1500 cm^−1^ region to show the MAH grafting onto PBS.

**Figure 5 molecules-26-05927-f005:**
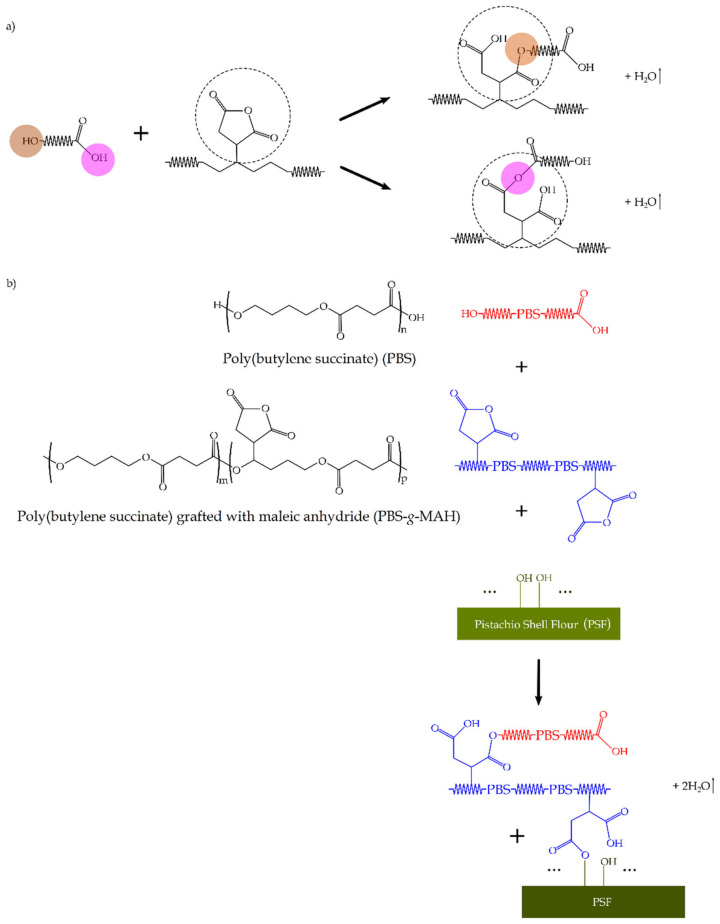
(**a**) Chemical reaction between the hydroxyl (−OH) end groups of poly(butylene succinate) (PBS) with maleic anhydride (MAH) groups; (**b**) Melt-grafting process of cellulose or lignin of pistachio shell flour (PSF) onto PBS by esterification reaction of their −OH groups with poly(butylene succinate) grafted with maleic anhydride (PBS-*g*-MAH).

**Figure 6 molecules-26-05927-f006:**
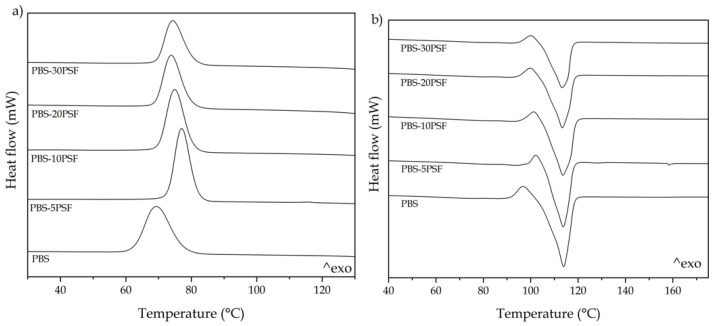
Differential scanning calorimetry (DSC) thermograms taken during cooling (**a**) and second heating (**b**) of the injection-molded pieces of poly(butylene succinate) (PBS)/pistachio shell flour (PSF) compatibilized with poly(butylene succinate) grafted with maleic anhydride (PBS-*g*-MAH).

**Figure 7 molecules-26-05927-f007:**
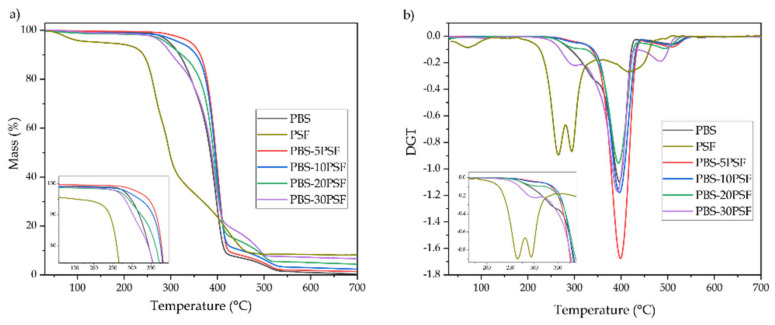
Thermogravimetric analysis (TGA) curves (**a**) and first derivate thermogravimetric (DTG) curves (**b**) of the injection-molded pieces of poly(butylene succinate) (PBS)/pistachio shell flour (PSF) compatibilized with poly(butylene succinate) grafted with maleic anhydride (PBS-*g*-MAH).

**Figure 8 molecules-26-05927-f008:**
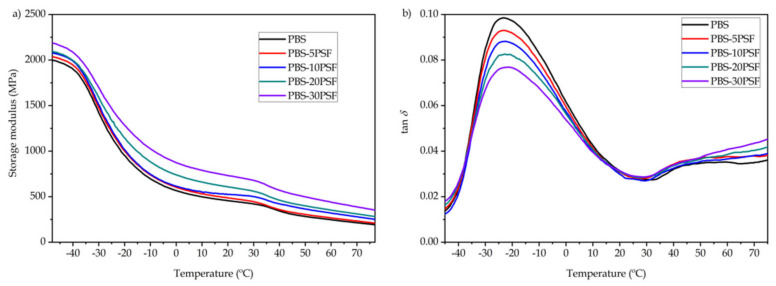
Evolution as function of the temperature of the storage modulus (**a**) and dynamic damping factor (*tan δ*) (**b**) of the injection-molded pieces of poly(butylene succinate) (PBS)/pistachio shell flour (PSF) compatibilized with poly(butylene succinate) grafted with maleic anhydride (PBS-*g*-MAH).

**Figure 9 molecules-26-05927-f009:**
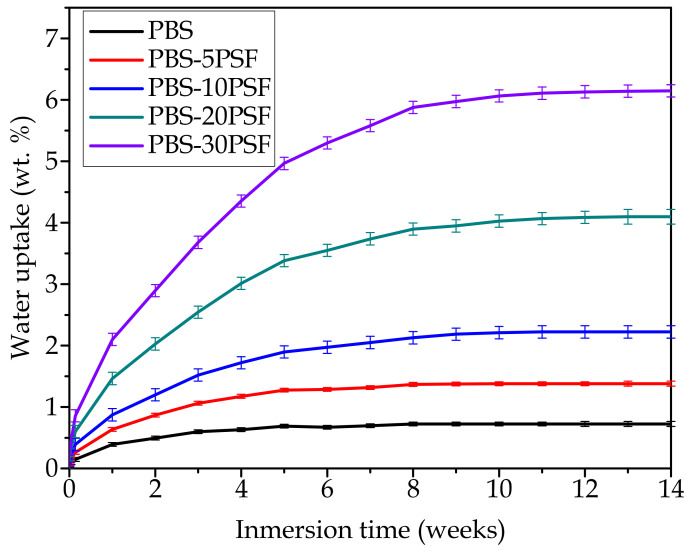
Evolution of the water uptake during a 14-week immersion period of the injection-molded pieces of poly(butylene succinate) (PBS)/pistachio shell flour (PSF) compatibilized with poly(butylene succinate) grafted with maleic anhydride (PBS-*g*-MAH).

**Figure 10 molecules-26-05927-f010:**
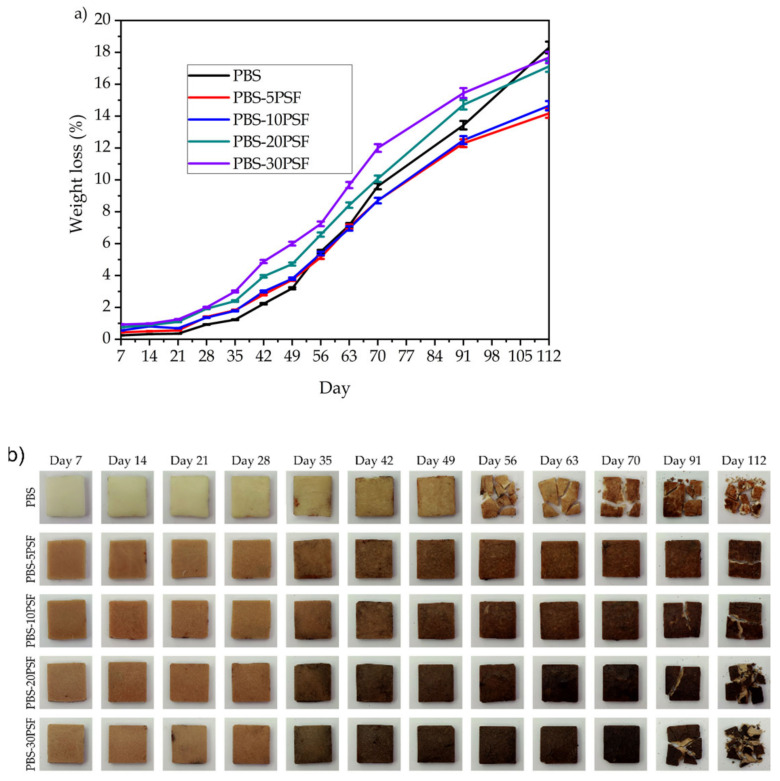
Weight loss as a function of elapsed time during disintegration test in controlled compost soil (**a**) and visual aspect (**b**) of the injection-molded pieces of poly(butylene succinate) (PBS)/pistachio shell flour (PSF) compatibilized with poly(butylene succinate) grafted with maleic anhydride (PBS-*g*-MAH).

**Figure 11 molecules-26-05927-f011:**
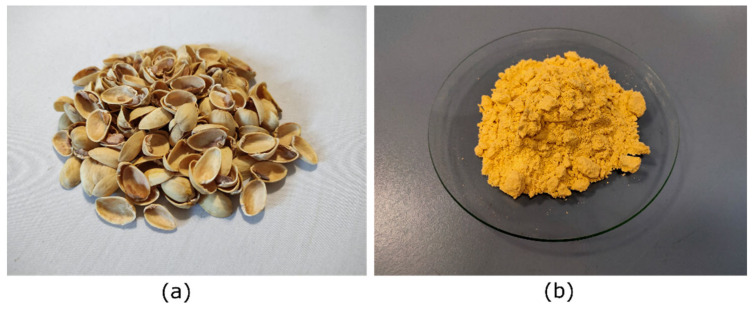
(**a**) Pistachio shells; (**b**) Resultant pistachio shell flour (PSF) after milling and sieving.

**Figure 12 molecules-26-05927-f012:**
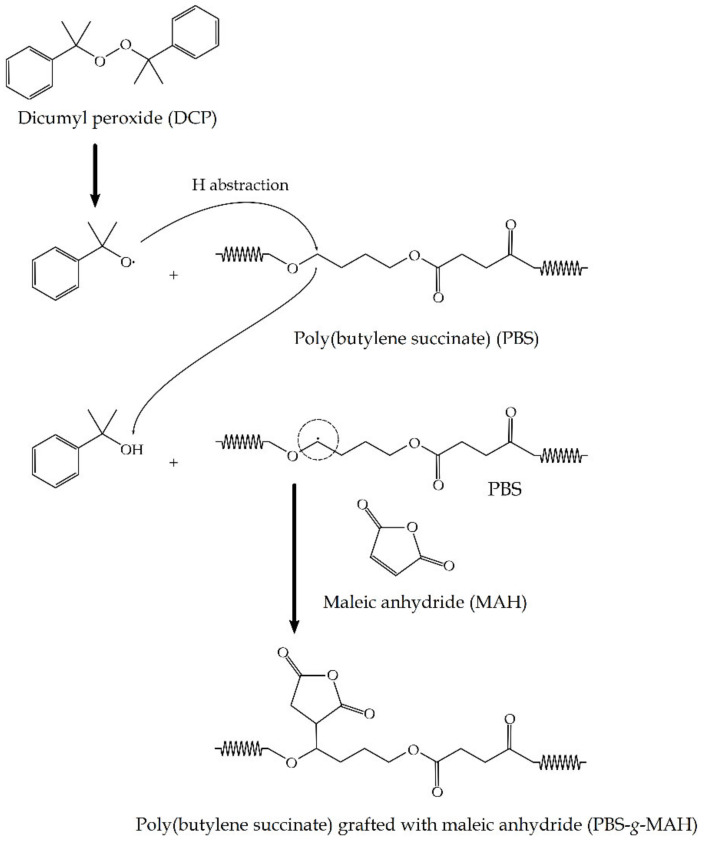
Reaction mechanism to obtain poly(butylene succinate) grafted with maleic anhydride (PBS-*g*-MAH) from poly(butylene succinate) (PBS) and maleic anhydride (MAH) induced by the presence of dicumyl peroxide (DCP).

**Table 1 molecules-26-05927-t001:** Luminance (L*), color coordinates (a*b*), and color difference ($\Delta E_{ab}^{*}$ ) of the injection-molded pieces of poly(butylene succinate) (PBS)/pistachio shell flour (PSF) compatibilized with poly(butylene succinate) grafted with maleic anhydride (PBS-*g*-MAH).

Piece	L*	a*	b*	$\Delta E_{ab}^{*}$
PBS	83.8 ± 0.1 ^a^	−1.8 ± 0.1 ^a^	1.1 ± 0.2 ^a^	-
PBS-5PSF	71.8 ± 0.2 ^b^	1.1 ± 0.1 ^b^	15.8 ± 0.2 ^b^	19.0 ± 0.5 ^a^
PBS-10PSF	67.6 ± 0.1 ^c^	2.8 ± 0.2 ^c^	20.1 ± 0.1 ^c^	25.0 ± 0.5 ^b^
PBS-20PSF	61.8 ± 0.1 ^d^	5.1 ± 0.2 ^d^	23.0 ± 0.1 ^d^	31.2 ± 0.7 ^c^
PBS-30PSF	57.8 ± 0.2 ^e^	6.8 ± 0.2 ^e^	25.1 ± 0.1 ^e^	35.7 ± 0.6 ^d^

^a–e^ Different letters in the same column indicate a significant difference among the samples (*p* < 0.05).

**Table 2 molecules-26-05927-t002:** Mechanical properties of the injection-molded pieces of poly(butylene succinate) (PBS)/pistachio shell flour (PSF) compatibilized with poly(butylene succinate) grafted with maleic anhydride (PBS-g-MAH) in terms of tensile modulus (E), maximum tensile strength (σ_max_), elongation at break (ε_b_), Shore D hardness, and impact strength.

Piece	E (MPa)	σ_max_ (MPa)	ε_b_ (%)	Shore D Hardness	Impact Strength (kJ/m^2^)
PBS	598.8 ± 13.4 ^a^	27.8 ± 0.5 ^a^	225.5 ± 15.2 ^a^	62.9 ± 0.7 ^a^	11.4 ± 0.8 ^a^
PBS-5PSF	605.6 ± 15.2 ^b^	23.1 ± 1.3 ^b^	42.9 ± 3.5 ^b^	63.6 ± 0.6 ^a^	6.8 ± 0.4 ^b^
PBS-10PSF	655.5 ± 18.7 ^c^	20.1 ± 0.8 ^c^	27.7 ± 1.7 ^c^	67.8 ± 0.4 ^b^	5.6 ± 0.2 ^c^
PBS-20PSF	852.2 ± 28.7 ^d^	17.0 ± 0.5 ^d^	15.9 ± 0.7 ^d^	68.8 ± 0.4 ^b^	3.3 ± 0.3 ^d^
PBS-30PSF	1039.6 ± 32.5 ^e^	16.0 ± 1.0 ^d^	9.7 ± 1.1 ^e^	69.5 ± 0.5 ^c^	2.9 ± 0.3 ^e^

^a–e^ Different letters in the same column indicate a significant difference among the samples (*p* < 0.05).

**Table 3 molecules-26-05927-t003:** Main thermal parameters of the injection-molded pieces of poly(butylene succinate) (PBS)/pistachio shell flour (PSF) compatibilized with poly(butylene succinate) grafted with maleic anhydride (PBS-*g*-MAH) in terms of crystallization temperature (T_c_), crystallization enthalpy (ΔH_c_), cold crystallization temperature (T_cc_), cold crystallization enthalpy (ΔH_cc_), melting temperature (T_m_), melting enthalpy (ΔH_m_), and crystallinity degree (X_c_).

Piece	T_c_ (°C)	ΔH_c_ (J/g)	T_cc_ (°C)	ΔH_cc_ (J/g)	T_m_ (°C)	ΔH_m_ (J/g)	X_c_ (%)
PBS	69.7 ± 0.9 ^a^	63.8 ± 0.8 ^a^	96.6 ± 0.4 ^a^	5.3 ± 0.8 ^a^	113.4 ± 0.8 ^a^	62.5 ± 0.9 ^a^	51.8 ± 1.2 ^a^
PBS-5PSF	77.7 ± 1.1 ^b^	65.9 ± 0.5 ^a^	102.0 ± 0.7 ^b^	3.0 ± 0.4 ^b^	113.3 ± 0.5 ^a^	57.3 ± 0.7 ^b^	52.2 ± 0.9 ^a^
PBS-10PSF	75.5 ± 0.7 ^c^	64.2 ± 1.0 ^a^	101.4 ± 0.6 ^b^	3.5 ± 0.6 ^c^	113.0 ± 0.7 ^a^	56.9 ± 1.1 ^b^	54.9 ± 1.0 ^b^
PBS-20PSF	74.3 ± 0.6 ^d^	53.8 ± 1.2 ^b^	99.7 ± 0.4 ^c^	3.5 ± 0.8 ^c^	112.8 ± 1.0 ^a^	56.6 ± 0.8 ^b^	62.8 ± 1.2 ^c^
PBS-30PSF	74.7 ± 1.1 ^d^	48.2 ± 0.5 ^c^	99.8 ± 0.9 ^c^	2.5 ± 0.5 ^d^	112.9 ± 0.6 ^a^	47.5 ± 1.2 ^c^	62.7 ± 0.7 ^c^

^a–d^ Different letters in the same column indicate a significant difference among the samples (*p* < 0.05).

**Table 4 molecules-26-05927-t004:** Main decomposition parameters of the injection-molded pieces of poly(butylene succinate) (PBS)/pistachio shell flour (PSF) compatibilized with poly(butylene succinate) grafted with maleic anhydride (PBS-*g*-MAH) in terms of onset degradation temperature at a mass loss of 5% (T_5%_), temperature of maximum degradation (T_deg_), and residual mass at 700 °C.

Piece	T_5%_ (°C)	T_max_ (°C)	Residual Mass (%)
PSF	224.8 ± 0.9 ^a^	264.8 ± 0.8 ^a^/293.7 ± 1.1 ^b^	8.2 ± 0.9 ^a^
PBS	296.6 ± 1.0 ^b^	398.3 ± 0.7 ^c^	0.4 ± 0.2 ^b^
PBS-5PSF	341.9 ± 0.8 ^c^	401.6 ± 1.5 ^c^	1.4 ± 0.5 ^c^
PBS-10PSF	318.3 ± 0.7 ^d^	399.0 ± 0.7 ^c^	2.3 ± 0.7 ^d^
PBS-20PSF	290.9 ± 1.2 ^e^	397.4 ± 0.8 ^c^	4.4 ± 0.5 ^e^
PBS-30PSF	279.6 ± 1.0 ^f^	394.9 ± 1.0 ^c^	6.6 ± 0.8 ^f^

^a–f^ Different letters in the same column indicate a significant difference among the samples (*p* < 0.05).

**Table 5 molecules-26-05927-t005:** Thermomechanical properties of the injection-molded pieces of poly(butylene succinate) (PBS)/pistachio shell flour (PSF) compatibilized with poly(butylene succinate) grafted with maleic anhydride (PBS-*g*-MAH) in terms of storage modulus measured at −45, 25, and 70 °C and dynamic damping factor (*tan δ*) peak.

Piece	Storage Modulus (MPa)			*tan δ* Peak (°C)
	−45 °C	25 °C	70 °C	
PBS	1973.3 ± 30 ^a^	440 ± 10 ^a^	214.3 ± 12 ^a^	−23.3 ± 0.2 ^a^
PBS-5PSF	2016.1 ± 35 ^a,b^	468 ± 21 ^a^	233.1 ± 20 ^a,b^	−23.0 ± 0.1 ^b^
PBS-10PSF	2056.5 ± 37 ^a,b^	516 ± 14 ^b^	279.9 ± 22 ^b,c^	−22.7 ± 0.3 ^c^
PBS-20PSF	2072.2 ± 38 ^a,b^	585 ± 12 ^c^	310.3 ± 17 ^c^	−22.3 ± 0.4 ^d^
PBS-30PSF	2161.1 ± 34 ^b^	705 ± 15 ^d^	387.0 ± 21 ^d^	−21.8 ± 0.2 ^e^

^a–e^ Different letters in the same column indicate a significant difference among the samples (*p* < 0.05).

**Table 6 molecules-26-05927-t006:** Summary of compositions according to the weight content (wt.%) of poly(butylene succinate) (PBS), pistachio shell flour (PSF), and poly(butylene succinate) grafted with maleic anhydride (PBS-*g*-MAH).

Composition	PBS (wt.%)	PSF (wt.%)	PBS-*g*-MAH (wt.%)
PBS	100.00	0.00	0.00
PBS-5PSF	94.20	5.00	0.83
PBS-10PSF	88.30	10.00	1.67
PBS-20PSF	76.60	20.00	3.30
PBS-30PSF	65.00	30.00	5.00

## Data Availability

Not applicable.

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
