# Peer review of "Peroxide-Induced Synthesis of Maleic Anhydride-Grafted Poly(butylene succinate) and Its Compatibilizing Effect on Poly(butylene succinate)/Pistachio Shell Flour Composites"

_molecules, 2021, doi:10.3390/molecules26195927_

Round 1

Reviewer 1 Report

The article deals with the secondary use of pistachio shell flour as a filler in biodegradable polybutylene succinate. To improve the compatibility of the two components, the authors prepare their own coupling agent by grafting of MAH on PBS in the melt state with Dicumylperoxide. Composites were produced in the range of 5-30% pistachio shell content, with relative same PBS-g-MAH : PSF amounts. The injection molded specimens are then extensively characterized (SEM, IR, DSC, TG, DMA, Tensile Mechanics, Water sorption, Degradation).

The strengths of the article lie in the comprehensive description of the composite, the weaknesses lie in the lack of discussion of materials theory regarding compatibilization. The synthesis of the compatibilizer and the mechanical data should be reconsidered in the light of material theories. There are doubts about the efficiency and effectiveness of the compatibilizer. The PBS-g-MAH should be checked for unreacted MAH and polymerized (MAH)n  or PBS-(MAH)n.

This work present experimental data in great detail showing the great deal of work that has been done, but in some passages, it seems as if the authors consistently focus the interpretation of the data on successful compatibilization.

Major und minor comments canbe find in the word file

Author Response

The strengths of the article lie in the comprehensive description of the composite, the weaknesses lie in the lack of discussion of materials theory regarding compatibilization. The synthesis of the compatibilizer and the mechanical data should be reconsidered in the light of material theories. There are doubts about the efficiency and effectiveness of the compatibilizer. The PBS-g-MAH should be checked for unreacted MAH and polymerized (MAH)n  or PBS-(MAH)n.

The MAH content on the synthetized PBS-g-MAH sample was determined. Please see new added information in section 4.2, page 20.

This work present experimental data in great detail showing the great deal of work that has been done, but in some passages, it seems as if the authors consistently focus the interpretation of the data on successful compatibilization.

The manuscript was modified to better demonstrate a succesful compatiblization.

Major und minor comments can be find in the word file.

Thank you for this intense and severe revision. I believe it has helped us to improve the manuscript. Please find attached the answer/comments for each comment.

Reviewer 2 Report

The manuscript is important for the green polymer composites. It could be published in the journal providing the following minor revisions:

1) It is important to give the evidence for the reaction between the mah groups with the reactive groups of the fillers. Please give a schematic diagram for that.

2) The notched impact strength is required.

3) The very recent literatures on the PBAT blends should be introduced in the introduction part: for example: Macromolecules 2021, 54, 2852-2861; Polymer 2020, 208, 122948etc

Author Response

1) It is important to give the evidence for the reaction between the mah groups with the reactive groups of the fillers. Please give a schematic diagram for that.

The schematic diagram shown in Figure 5 has been improved to show the reaction between the multiple MAH groups present in PBS-g-MAH with both PBS and PSF.

2) The notched impact strength is required.

Impact strength was measured using V-notched samples with a radius of 0.25 mm. Please see details in section 4.4.2, page 22.

3) The very recent literatures on the PBAT blends should be introduced in the introduction part: for example: Macromolecules 2021, 54, 2852-2861; Polymer 2020, 208, 122948etc

The recent references were added and discussed in the Introduction. Please see lines 124-126.